# DynamicVAE: Decoupling Reconstruction Error and Disentangled Representation Learning

## Abstract

This paper challenges the common assumption that the weight $\beta$, in $\beta$-VAE, should be larger than 1 in order to effectively disentangle latent factors. We demonstrate that $\beta$-VAE, with $\beta < 1$, can not only attain good disentanglement but also significantly improve reconstruction accuracy via dynamic control. The paper *removes the inherent trade-off* between reconstruction accuracy and disentanglement for $\beta$-VAE. Existing methods, such as $\beta$-VAE and FactorVAE, assign a large weight to the KL-divergence term in the objective function, leading to high reconstruction errors for the sake of better disentanglement. To mitigate this problem, a ControlVAE has recently been developed that dynamically tunes the KL-divergence weight in an attempt to *control the trade-off* to more a favorable point. However, ControlVAE fails to eliminate the conflict between the need for a large $\beta$ (for disentanglement) and the need for a small $\beta$ (for smaller reconstruction error). Instead, we propose DynamicVAE that maintains a different $\beta$ at different stages of training, thereby *decoupling disentanglement and reconstruction accuracy*. In order to evolve the weight, $\beta$, along a trajectory that enables such decoupling, DynamicVAE leverages a modified incremental PI (proportional-integral) controller, a variant of proportional-integral-derivative controller (PID) algorithm, and employs a moving average as well as a hybrid annealing method to evolve the value of KL-divergence smoothly in a tightly controlled fashion. We theoretically prove the stability of the proposed approach. Evaluation results on three benchmark datasets demonstrate that DynamicVAE significantly improves the reconstruction accuracy while achieving disentanglement comparable to the best of existing methods. The results verify that our method can separate disentangled representation learning and reconstruction, removing the inherent tension between the two.

## 1 Introduction

The goal of disentangled representation learning is to encode input data into a low-dimensional space that preserves information about the salient factors of variation, so that each dimension of the representation corresponds to a distinct factor in the data (Bengio et al., 2013; Locatello et al., 2020; van Steenkiste et al., 2019). Learning disentangled representations benefits a variety of downstream tasks (Higgins et al., 2018; Lake et al., 2017; Locatello et al., 2019c;a; Denton et al., 2017; Mathieu et al., 2019), including abstract visual reasoning (van Steenkiste et al., 2019), zero-shot transfer learning (Burgess et al., 2018; Lake et al., 2017; Higgins et al., 2017a) and image generation (Nie et al., 2020), just to name a few. Due to its central importance in various downstream applications, there is abundant literature on learning disentangled representations. Roughly speaking, there are two lines of methods towards this goal. The first category includes supervised methods (Chen & Batmanghelich, 2019; Locatello et al., 2019c; Shu et al., 2019; Bouchacourt et al., 2018; Nie et al., 2020; Yang et al., 2015), where external supervision (e.g., data generative factors) is available during training to guide the learning of disentangled representations. The second line of works focus on unsupervised methods (Chen et al., 2016; 2018; Burgess et al., 2018; Kim & Mnih, 2018; Denton et al., 2017; Kumar et al., 2018; Fraccaro et al., 2017), which substantially relieve the needs to have external supervisions. For this reason, in this paper, we mainly focus on *unsupervised* disentangled representation learning.

One major challenge of unsupervised disentanglement learning is that there exists a trade-off between reconstruction quality of the input signal and the degree of disentanglement in the latent representations. Let us take $\beta$-VAE and its variants (Burgess et al., 2018; Chen et al., 2018; Higgins et al., 2017a) as an example. These methods assign a large and fixed weight $\beta$ in the objective function to improve the disentanglement at the cost of reconstruction quality, which is highly correlated with accuracy in downstream tasks (van Steenkiste et al., 2019; Locatello et al., 2020). In order to improve the reconstruction quality, researchers have proposed a dynamic learning approach, ControlVAE (Shao et al., 2020), to dynamically adjust the weight on the KL term in the VAE objective to better balance the quality of disentangled representation learning and reconstruction error. However, while ControlVAE allows better control of the trade-off between disentangled representation learning and reconstruction error, it *does not eliminate it*. One is still achieved at the expense of the other. The contribution of this paper, compared to the above state of the art, lies in demonstrating that with the proper design, the trade-off between disentangled representation learning and reconstruction error is completely eliminated. Both objectives can be attained at the same time in a decoupled fashion, without affecting each other. More specifically, we observe that if $\beta$ was kept high in the beginning of training then lowered later in the process, the two objectives are decoupled allowing each to be independently optimized. To the authors' knowledge, this work is the first to attain such decoupled optimization of both quality of disentanglement and reconstruction error.

**Our Contributions:**   In this paper, we propose a novel unsupervised disentangled representation learning method, dubbed as DynamicVAE, that turns the weight of $\beta$-VAE ($\beta > 1$) (Burgess et al., 2018; Higgins et al., 2017a) into a small value ($\beta \leq 1$) to achieve not only good disentanglement but also a high reconstruction accuracy via dynamic control. We summarize the main contributions of this paper as follows.

- We propose a new model, DynamicVAE, that leverages an incremental PI controller and moving average to *evolve the desired KL-divergence along a trajectory that enables decoupling of two objectives*: high-quality disentanglement and low reconstruction error.
- We provide the theoretical conditions on parameters of the PI controller to guarantee stability of DynamicVAE.
- We experimentally demonstrate that our approach turns the weight of $\beta$-VAE ($\beta > 1$) to $\beta \leq 1$, achieving higher reconstruction quality yet comparable disentanglement compared to prior approaches (e.g., FactorVAE). Thus, our results verify that the proposed method indeed decouples disentanglement and reconstruction accuracy without hurting each other's performance.

## 2 PRELIMINARIES

$\beta$**-VAE and its Variants:**  $\beta$-VAE (Higgins et al., 2017b; Chen et al., 2018) is a popular unsupervised method for learning disentangled representations of the data generative factors (Bengio et al., 2013). Compared to the original VAE, $\beta$-VAE incorporates an extra hyperparameter $\beta(\beta > 1)$ as the weight of the KL term in the VAE objective:

$$\mathcal{L}_\beta = \mathbb{E}_{q_\phi(\mathbf{z}|\mathbf{x})}[\log p_\theta(\mathbf{x}|\mathbf{z})] - \beta D_{KL}(q_\phi(\mathbf{z}|\mathbf{x}) \parallel p(\mathbf{z})). \tag{1}$$

In order to discover more disentangled factors, in other variants, practitioners further add a constraint on the total information capacity, $C$, to control the capacity of the latent channels (Burgess et al., 2018) to transmit information. The constraint can be formulated as an optimization method:

$$\mathcal{L}_\beta = \mathbb{E}_{q_\phi(\mathbf{z}|\mathbf{x})}[\log p_\theta(\mathbf{x}|\mathbf{z})] - \beta \cdot |D_{KL}(q_\phi(\mathbf{z}|\mathbf{x})\|p(\mathbf{z})) - C|, \tag{2}$$

where $\beta$ is a large and fixed hyperparameter. As a result, when the weight $\beta$ is large (e.g. 100), the algorithm tends to optimize the second term in (2), leading to much higher reconstruction error.

**PID Control Algorithm:**   The PID is a simple yet effective control algorithm that can stabilize system output to a desired value via feedback control (Stooke et al., 2020; Åström et al., 2006). The PID algorithm calculates an error, $e(t)$, between a set point (in this case, the desired KL-divergence) and the current value of the controlled variable (in this case, the actual KL-divergence), then applies a correction in a direction that reduces that error. The correction is the weighted sum of three terms, one *proportional* to the error (called P), one that is the *integral* of error (called I), and one that is the *derivative* of error (called D); thus, the term PID. The derivative term is not recommended for noisy systems, such as ours, reducing the algorithm to PI control. The canonical form of a PI controller

(applied to control $\beta(t)$) is the following:

$$\beta(t) = K_p e(t) + K_i \sum_{j=0}^{t} e(j), \tag{3}$$

where $\beta(t)$ is the output of a controller, which (in our case) is the used $\beta$ during training at time $t$; $e(t)$ is the error between the output value and the desired value at time $t$; $K_p, K_i$ denote the coefficients for the P term and I term, respectively. Eq. (3) may be rewritten in incremental form, as follows:

$$\beta(t) = \Delta\beta(t) + \beta(t-1), \tag{4}$$

where $\beta(0)$ can be set as needed (as we show later), and:

$$\Delta\beta(t) = K_p[e(t) - e(t-1)] + K_i e(t). \tag{5}$$

This paper adopts a nonlinear incremental form of the PI controller, described later in Section 3.

## 3 THE DYNAMICVAE ALGORITHM

The goal of disentangled representation learning (Burgess et al., 2018) is to maximize the log likelihood and simultaneously stabilize the KL-divergence to a target value $C$. It can be formulated as the following constrained optimization problem:

$$\max_{\phi,\theta} \quad \mathbb{E}_{q_\phi(\mathbf{z}|\mathbf{x})}[\log p_\theta(\mathbf{x}|\mathbf{z})], \qquad \text{s.t.} \quad D_{KL}(q_\phi(\mathbf{z}|\mathbf{x}) \,\|\, p(\mathbf{z})) = C \tag{6}$$

In order to achieve a good trade-off between disentanglement and reconstruction accuracy, we attempt to design a controller to dynamically adjust $\beta(t)$ in the following VAE objective to stabilize the KL-divergence to the desired value $C$:

$$\mathcal{L}_d = \mathbb{E}_{q_\phi(\mathbf{z}|\mathbf{x})}[\log p_\theta(\mathbf{x}|\mathbf{z})] - \beta(t) D_{KL}(q_\phi(\mathbf{z}|\mathbf{x}) \,\|\, p(\mathbf{z})). \tag{7}$$

The contribution of DynamicVAE is to evolve $\beta(t)$ along a good trajectory to achieve decoupling between disentanglement and reconstruction error.

To reach this goal, we need to address the following two challenges:

1. $\beta(t)$ should dynamically change from a large value to small one. Specifically, at the beginning of training, $\beta(t)$ should be large enough to disentangle latent factors. After that, $\beta(t)$ is required to gradually drop to a small value to optimize the reconstruction.
2. $\beta(t)$ should not change too fast or oscillates too frequently. When $\beta(t)$ drops too fast or oscillates, it may cause KL-divergence to grow with a large value. Consequently, some latent factors may emerge earlier so that they can potentially be entangled with each other.

In this paper, we propose methods to deal with these two challenges, summarized below.

**A non-linear incremental PI controller:** Fig. 1 (a) shows the designed non-linear PI controller that dynamically adjusts the weight $\beta(t)$ in the KL term of the $\beta$-VAE, based on the actual KL-divergence, $y_{KL}(t)$. Specifically, it first samples the output KL-divergence, $y_{KL}(t)$, at training step $t$. Then we use the difference $e(t)$ between the sampled KL-divergence at time $t$ with the desired value, $C$, as the feedback to PI controller to tune $\beta(t)$. The corresponding PI algorithm is denoted by

$$\beta(t) = K_p \sigma(-e(t)) - K_i \sum_{j=0}^{t} e(j), \tag{8}$$

where $\sigma(.)$ is a sigmoid function; $K_p, K_i$ are positive hyper-parameters for P and I term respectively. As mentioned earlier, we need a large $\beta(t)$ in the beginning to control the KL-divergence from a small value to a large target value so that the information can be transmitted through the latent channels per data sample. Accordingly, we adopt an incremental form of the PI controller in Eq. (8), and initialize it to a large value:

$$\beta(t) = \Delta\beta(t) + \beta(t-1), \tag{9}$$

where

$$\Delta\beta(t) = K_p[\sigma(-e(t)) - \sigma(-e(t-1))] - K_i e(t). \tag{10}$$

and $\beta(0)$ is a large initial value. When the PI controller is initialized to a large value $\beta(0)$, it can quickly produce a (small) KL-divergence during initial model training, preventing emergence of entangled factors.

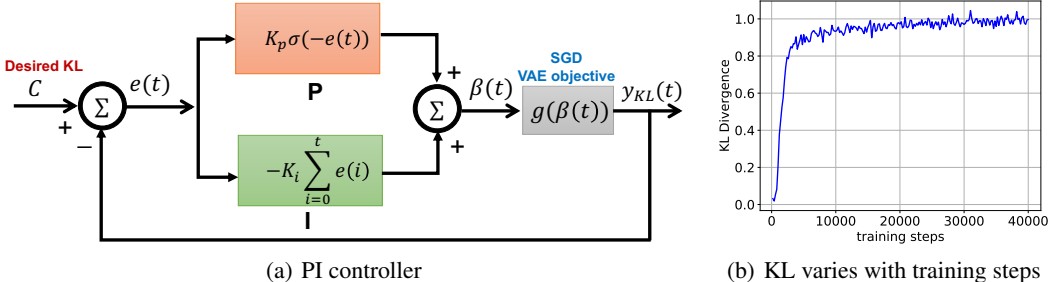

(a) PI controller

(b) KL varies with training steps

Figure 1: (a) PI control algorithm. (b) Time response of KL-divergence when $\beta = 130$.

**Moving average:** Since our model is trained with mini-batch data, it often contains noise that causes $\beta(t)$ to oscillate. In particular, when $\beta(t)$ plunges during training, it would cause KL-divergence to rise too quickly. This may lead to multiple latent factors coming out together to be entangled. To mitigate this issue, we adopt moving average to smooth the output KL-divergence as the feedback of PI controller below.

$$y(t) = \alpha_t y_{KL}(t) + \alpha_{t-1} y_{KL}(t-1) + \cdots + \alpha_{t-T} y_{KL}(t-T) = \sum_{i=t-T}^{t} \alpha_i y_{KL}(i), \qquad (11)$$

where $\alpha_i$ denotes weight and $T$ denotes the window size of past training steps.

**Hybrid annealing:** Control systems with step (input) function (i.e., those where the set point can change abruptly) often suffer from an overshoot problem. An overshoot is temporary overcompensation, where the controlled variable oscillates around the set point. In our case, it means that the actual KL-divergence may significantly (albeit temporarily) exceed the desired value, when set point is abruptly changed. This effect would cause some latent factors to come out earlier than expected, and be entangled, thereby producing poor-quality disentanglement. To address this problem, we develop a hybrid annealing method that changes the set point more gradually, as illustrated in Fig. 7 in Appendix. It combines step function with ramp function to smoothly increase the target KL-divergence in order to prevent overshoot and thus better disentangle latent factors one by one.

The combination of the above three methods allows DynamicVAE to evolve $\beta(t)$ along a favorable trajectory to separate disentanglement learning and reconstruction optimization. We summarize the proposed incremental PI algorithm in Algorithm 1 in Appendix B.

### 3.1 STABILITY ANALYSIS OF DYNAMICVAE

We further theoretically analyze the stability of the proposed DynamicVAE. This work is the first to offer the necessary and sufficient conditions that control hyperparameters should satisfy in order to guarantee the stability of KL-divergence, when $\beta$ is manipulated dynamically during the training process of a (variant of) $\beta$-VAE.

To this end, our first step is to build the state space model for our control system. Throughout the paper, the state variable at training step $t$ is defined as $x(t) = \beta(t)$. Accordingly, the model of incremental PI controller can be written as:

$$x(t+1) - x(t) = K_p[\sigma(-e(t)) - \sigma(-e(t-1))] - K_i e(t), \qquad (12)$$

where error $e(t)$, as shown in Fig. 1(a), is given by $e(t) = C - y(t-1)$. Here $y(t)$ is a dynamic model about the time response of the output KL divergence, $y_{KL}(t)$. According to Liu & Theodorou (2019), stochastic gradient descent (SGD) for optimizing an objective function can be described by a first-order dynamic model. Our experiment, as illustrated in Fig. 1(b), also shows that the response $y(t)$ in the open loop system approximately meets a negative exponential function, which further verifies that our system is a first-order dynamic system. We hence use the first-order dynamic model to describe it below.

$$\frac{dy}{dt} + ay = ag(x), \qquad (13)$$

where $a$ is a positive hyperparameter to describe the dynamic property, and $g(x)$ is a mapping function between the actual KL-divergence and $\beta(t)$. Since DynamicVAE is a discrete control system with

sampling period $T_s = 1$, the above first-order dynamic model can be reformulated as

$$y(t) - y(t-1) + ay(t) = ag(x(t)) \implies y(t) = \frac{1}{1+a}y(t-1) + \frac{a}{1+a}g(x(t)). \quad (14)$$

Now let $x_1(t) = x(t), x_2(t) = y(t-1), x_3(t) = y(t-2)$, then Eqs. (12) and (14) can be rewritten as the following state space equations.

$$\begin{cases} x_1(t+1) = x_1(t) - K_i[C - x_2(t)] + K_p[\sigma(x_2(t) - C) - \sigma(x_3(t) - C)] \triangleq f_1(x_1(t), x_2(t), x_3(t)) \\ x_2(t+1) = \frac{a}{1+a}g(x_1(t)) + \frac{1}{1+a}x_2(t) \triangleq f_2(x_1(t), x_2(t), x_3(t)) \\ x_3(t+1) = x_2(t) \triangleq f_3(x_1(t), x_2(t), x_3(t)). \end{cases}$$
$$(15)$$

In order to analyze the stability of the above non-linear state space model, one commonly used method is to linearize it at an equilibrium point (Hughes, 2015). In this paper, we use the following equilibrium point:

$$x^* = (x_1^*, x_2^*, x_3^*) = (g^{-1}(C), C, C), \quad (16)$$

where $g^{-1}(\cdot)$ denotes the inverse function and $x_2^* = x_3^*$. Next, we apply the first-order Taylor expansion to the above Eq. (15), yielding

$$X(t+1) = AX(t), \quad (17)$$

where

$$X(t) = [x_1(t) - x_1^*, x_2(t) - x_2^*, x_3(t) - x_3^*]^T, \quad (18)$$

and $A$ is the Jacobian matrix at equilibrium point $x^*$, as defined in Eq. (20) in Appendix C. After this linearization, we can prove the stability of the proposed method as the modulus of eigenvalue $\lambda$ of $A$ is smaller than 1, as described in the following theorem.

**Theorem 3.1.** Let $a > 0$ and assume $g'(x) < 0, \forall x > 0$. Then DynamicVAE is stable at the equilibrium point $C$ if and only if the parameters of the PI controller, $K_i$ and $K_p$, satisfy the following conditions

$$\begin{cases} K_p + K_i < -\frac{4(1+a)}{ag'(x_1^*)} \\ -0.5K_p^2 ag'(x_1^*)^2 - 2[K_p - 8K_i(1+a)]g'(x_1^*) + 8(1+a) > 0 \\ K_i > 0, K_p > 0 \end{cases} \quad (19)$$

*We provide the detailed proof in Appendix C.*

**Remark 3.1.** The assumption of $g'(x) < 0, \forall x$ basically asks that the KL term in the objective to be a monotonously decreasing function of the coefficient $\beta(t)$, and we also further empirically corroborate its validity on two benchmark datasets as shown in Appendix C.1. In addition, we choose $K_p$ and $K_i$ that meet the above conditions (19) to verify the stability of DynamicVAE in Appendix C.1.

## 4 EXPERIMENTS

In this section we evaluate the performance of DynamicVAE and compare it against existing baselines, including ControlVAE (Shao et al., 2020), $\beta$-VAE$_H$ (Higgins et al., 2017b), $\beta$-VAE$_B$ (Burgess et al., 2018), FactorVAE (Kim & Mnih, 2018), and VAE (Kingma & Welling, 2013). We conduct experiments on three benchmark datasets: dSprites (Burgess et al., 2018), MNIST (Chen et al., 2016) and 3D Chairs (Aubry et al., 2014). The detailed model configurations and hyperparameter settings are presented in Appendix A. Source code will be publicly available upon publication.

### 4.1 RESULTS AND ANALYSIS

**Dsprites Dataset:** We first evaluate the performance of DynamicVAE on learning disentangled representations using *dSprites*. Fig. 2 (a) and (b) illustrate the comparison of reconstruction error and the hyperparameter $\beta(t)$ (using 5 random seeds) for different approaches. We can observe from Fig. 2 (a) that DynamicVAE (KL=20) has much lower reconstruction error (about 11.8) than $\beta$-VAE and FactorVAE, and is comparable to the basic VAE and ControlVAE. This is because DynamicVAE dynamically adjusts the weight, $\beta(t)$, to balance the disentanglement and reconstruction. Specifically,

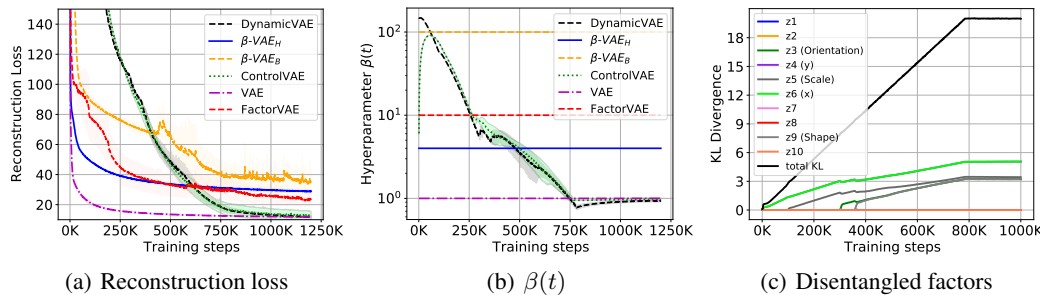

(a) Reconstruction loss      (b) $\beta(t)$      (c) Disentangled factors

Figure 2: (a) shows the comparison of reconstruction error on dSprites using 5 random seeds. DynamicVAE (KL=20) has comparable reconstruction errors as the basic VAE. (b) shows that DynamicVAE turns the weight of $\beta$-VAE into a small value less than 1. (c) shows an example of DynamicVAE on disentangling factors as the total KL-divergence increases.

Table 1: RMIG for different methods averaged over 5 random seeds. The higher the better.

| Models/Metric | pos. $x$ | pos. $y$ | Shape | Scale | Orientation | RMIG |
|---|---|---|---|---|---|---|
| DynamicVAE (KL=20) | 0.7166 | 0.7179 | 0.2004 | 0.6530 | 0.1024 | **0.4781 $\pm$ 0.0172** |
| ControlVAE (KL=20) | 0.6802 | 0.6597 | 0.0956 | 0.6040 | 0.1081 | 0.4295 $\pm$ 0.0865 |
| FactorVAE ($\gamma = 10$) | 0.7482 | 0.7276 | 0.1383 | 0.6262 | 0.1412 | 0.4763 $\pm$ 0.0513 |
| $\beta$-VAE$_B$ ($\gamma = 100$) | 0.5666 | 0.5763 | 0.4353 | 0.3814 | 0.0631 | 0.4045 $\pm$ 0.0345 |
| $\beta$-VAE$_H$ ($\beta = 4$) | 0.1635 | 0.1047 | 0.1391 | 0.3958 | 0.0127 | 0.1632 $\pm$ 0.0626 |
| VAE | 0.0359 | 0.0243 | 0.0116 | 0.1507 | 0.0039 | 0.0452 $\pm$ 0.0326 |

DynamicVAE automatically assigns a large $\beta(t)$ at the beginning of training in order to obtain good disentanglement, and then its weight gradually drops to less than 1 at the end of optimization, as shown in Fig. 2 (b). In contrast, $\beta$-VAE and FactorVAE have a large and fixed weight in the objective so that their optimization algorithms tend to optimize the KL-divergence term (total correlation term for FactorVAE), leading to higher reconstruction error. For ControlVAE, it can also dynamically tune $\beta(t)$ to control the value of KL-divergence, but its disentanglement performance degrades with the increase of KL-divergence (i.e., decrease of weight) as illustrated in Table 1. In addition, Fig. 2(c) illustrates an example of per-factor KL-divergence in the latent code as the total information capacity (KL-divergence) increases from 0.5 to 20. We can see that DynamicVAE disentangles all the five data generative factors, starting from position ($x$ and $y$) to scale, followed by orientation and shape.

Next, we use a robust disentanglement metric, robust mutual information gap (RMIG) (Do & Tran, 2020), to evaluate the disentanglement of different methods. We can see from Table 1 that DynamicVAE has a comparable RMIG score to the FactorVAE, but it has much lower reconstruction error as illustrated in Fig. 2. Moreover, DynamicVAE has higher RMIG score but lower reconstruction error than $\beta$-VAE models. We also find that our method achieves much better disentanglement than ControlVAE for comparable reconstruction accuracy. Hence, DynamicVAE is able to improve the reconstruction quality yet obtain good disentanglement.

Qualitatively, we also visualize the disentanglement results of different models in Fig. 3. We can observe that DynamicVAE disentangles all the five generative factors on dSprites. However, ControlVAE is not very effective to disentangle all the factors when its KL-divergence is set to a large value, such as 20. Furthermore, $\beta$-VAE$_B$ ($\gamma = 100$) disentangles four generative factors and mistakenly combines the scale and shape factors together (in the third row). The other methods do not perform well for disentanglement.

**MNIST and 3D Chair Datasets** We also evaluate the proposed method on the other two datasets: MNIST and 3D Chairs. Fig. 4 illustrates some samples of the disentangled factors for DynamicVAE on MNIST. We also verify that our method achieves better disentanglement compared with the other methods, shown in Appendix D. In addition, our method with $\beta < 1$ can significantly improve the reconstruction accuracy than $\beta$-VAE as illustrated in Fig. 10 in Appendix D.

We also demonstrate that DynamicVAE can learn many different data generative factors on another challenging dataset, 3D Chairs. We can observe from Fig. 5 that our method disentangles six different latent factors, such as wheels, and leg height and azimuth, same as FactorVAE (Kim & Mnih, 2018).

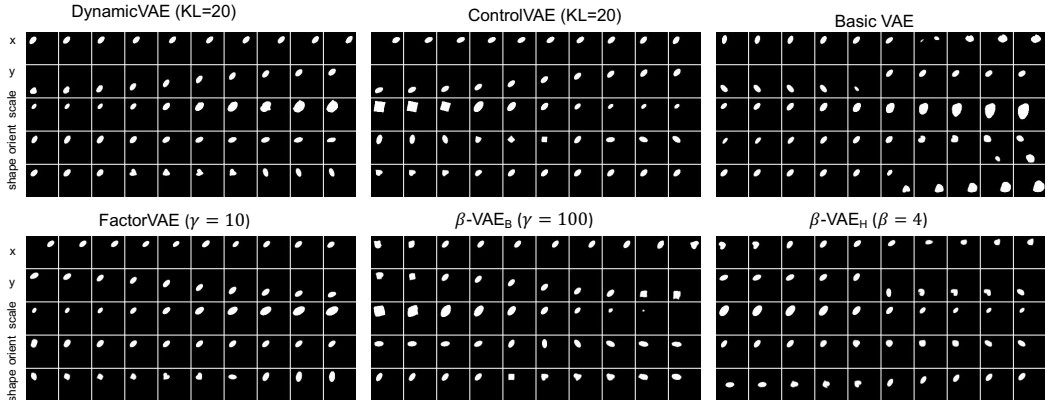

Figure 3: Rows: latent traversals ordered by the value of KL-divergence with the prior in a descending order. We initialize the latent representation from a seed image, and then traverse a single latent code in a range of $[-3, 3]$, while keeping the remaining latent code fixed.

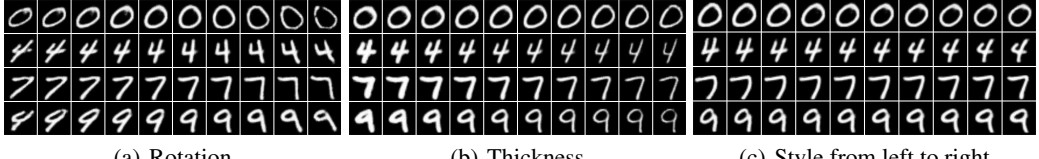

| (a) Rotation | (b) Thickness | (c) Style from left to right |

Figure 4: Latent traversals on MNIST for DynamicVAE. It can be seen that our method can disentangle four different factors: rotation, thickness, size(width) and writing style.

**Decoupled Reconstruction and Disentanglement** Additionally, we show that the proposed DynamicVAE is able to decouple the reconstruction and disentanglement learning into two phases, overcoming the problem of balancing the trade-off between reconstruction and disentanglement. Fig. 6 illustrates the RMIG score and reconstruction loss with the increase of training steps after all the factors are disentangled (before $800,000$). It can be seen that RMIG score of our method remains stable as the reconstruction loss drops. Therefore, the proposed method does not introduce any conflict between reconstruction optimization and disentangled representation learning.

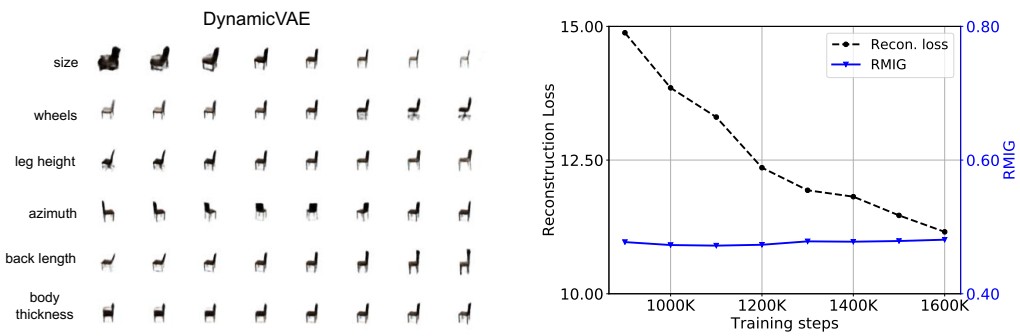

Figure 5: Sample traversals for the six latent factors in our model on 3D Chairs.

Figure 6: Averaged RMIG score and reconstruction loss vary with training steps.

### 4.2 ABLATION STUDIES

To compare the performance of DynamicVAE and its variants, we perform following ablation studies:

- DynamicVAE-P: it uses positional PI controller with no initialization to a large $\beta(0)$, instead of the incremental PI initialized to a large $\beta(0)$, to tune the weight on KL term in the VAE objective.
- DynamicVAE-step: it solely adopts step function without ramp function for our annealing method.
- DynamicVAE-t: this model directly uses the output KL-divergence at time $t$ as a feedback of PI controller without using moving average to smooth it.

Table 2: RMIG for different methods averaged over 5 random seeds. The higher is better.

| Models/Metric | pos. $x$ | pos. $y$ | Shape | Scale | Orientation | RMIG |
|---|---|---|---|---|---|---|
| DynamicVAE | 0.7166 | 0.7179 | 0.2004 | 0.6530 | 0.1024 | **0.4781** $\pm$ 0.0172 |
| DynamicVAE-P | 0.7376 | 0.7317 | 0.0992 | 0.6400 | 0.1120 | 0.4641 $\pm$ 0.0240 |
| DynamicVAE-step | 0.7209 | 0.7143 | 0.0664 | 0.6218 | 0.1543 | 0.4555 $\pm$ 0.0355 |
| DynamicVAE-t | 0.7152 | 0.7110 | 0.0997 | 0.6267 | 0.1322 | 0.4570 $\pm$ 0.0182 |

Table 2 shows the comparison of RMIG score for DynamicVAE and its variants. It can be observed that DynamicVAE outperforms the other methods in terms of overall RMIG score. We also find that DynamicVAE-step does not perform well because the ramp function is removed from our annealing method, leading to overshoot of PI controller. As a result, it makes the other factors come out earlier and entangled to each other. Thus, we can conclude the importance of adding ramp function for our annealing method. In addition, we can see that the proposed moving average and incremental PI control algorithm also play a critical role to improve the disentanglement.

## 5 RELATED WORK

Disentangled representation learning can be divided into two main categories: unsupervised learning and supervised learning.

Supervised disentanglement learning (Mathieu et al., 2016; Siddharth et al., 2017; Kingma et al., 2014; Reed et al., 2014) requires the prior knowledge of some data generative factors from human annotation to train the model. Some studies (Locatello et al., 2019a;b;c) figure out that it is hard to achieve reliable and good disentanglement without supervision. For supervised learning, the limited labeling information can help ensure a latent space of the VAE with desirable structure w.r.t to the ground-truth latent factors. In order to reduce human annotations, researchers tried to develop weakly supervised learning (Bouchacourt et al., 2018; Hosoya, 2019; Locatello et al., 2020) to learn disentangled representations. However, these methods still require explicit human labeling or assume the change of the two observations is small. In practice, it is unrealistic for initial learners to discover the data generative factors in most real world scenarios.

For unsupervised learning methods, the recent approaches mainly build on Variational Autoencoders (VAEs) (Kingma & Welling, 2013) and Generative Adversarial Networks (GANs) (Goodfellow et al., 2014). InfoGAN (Lin et al., 2020) is the first scalable unsupervised learning method for disentangling. It, however, suffers from training instabilities and does not perform well in disentanglement learning (Higgins et al., 2018), so most recent works are largely based on VAEs models. The VAE models, such as $\beta$-VAE ($\beta > 1$), FactorVAE and $\beta$-TCVAE (Chen et al., 2018), often suffer from high reconstruction errors in order to obtain better disentanglement, since they add a large weight to terms in the objective. To address this issue, Shao et al. (2020) develops a controllable variational autoencoder (ControlVAE) to dynamically tune $\beta$ to achieve the trade-off between reconstruction quality and disentanglement. However, ControlVAE fails to decouple disentanglement and reconstruction accuracy because the computed $\beta$ may oscillate during training, since it uses step function as annealing method and frequently adjusts $\beta$ at each step. In this paper, we propose a new method, DynamicVAE, that can separate disentangled representation learning and reconstruction.

## 6 CONCLUSION

This paper proposed a novel dynamic learning method, DynamicVAE, to address the trade-off problem between reconstruction and disentanglement. Our method is able to turn the weight of $\beta$-VAE to a small value ($\beta \leq 1$) to achieve good disentanglement against the previous default consumption, $\beta > 1$. Specifically, we leveraged an incremental PI controller, moving average and a hybrid annealing to stabilize the KL-divergence to decouple the disentanglement and reconstruction. We further theoretically prove the stability of the proposed DynamicVAE. The evaluation results demonstrate DynamicVAE can significantly improve the reconstruction accuracy meanwhile attaining good disentanglement. It decouples disentanglement and reconstruction accuracy without introducing a conflict between them.

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

## A    MODEL CONFIGURATIONS AND HYPERPARAMETER SETTINGS

We summarize the detailed model configurations and hyperparameter settings for DynamicVAE below.

Following the same model architecture of $\beta$-VAE, we adopt a convolutional layer and deconvolutional layer for our experiments. We use Adam optimizer with $\beta_1 = 0.90$, $\beta_2 = 0.99$ and a learning rate tuned from $10^{-4}$. We set $K_p$ and $K_i$ for PI algorithm to 0.01 and 0.005, respectively. The weight $\beta(t)$ for incremental PI controller is initialized with 150, 100 and 50 for dSprites, MNIST and 3D Chairs, respectively. The batch size is set to 128. Using the similar methodology in (Burgess et al., 2018), we train a single model by gradually increasing KL-divergence from 0.5 to a desired value $C$ with a step function $s$ and ramp function for every $M$ training steps, as shown in Fig. 7. In the experiment, we set the step, $s$, to 0.15 per $M = 6,000$ training steps (including $5,000$ in step function and $1,000$ in ramp function) as the information capacity (desired KL- divergence) increases from 0.5 until to 20, 26 and 18 for dSprites, MNIST and 3D Chairs datasets respectively. In addition, the window size of moving average is $T = 5$ with equal weight $\alpha$. Our model adopts the same encoder and decoder architecture as $\beta$-VAE$_H$ and ControlVAE except for plugging in PI control algorithm, as illustrated in Table 3 and Table 4.

Table 3: Encoder and decoder architecture for disentangled representation learning on dSprites and MNIST.

| Encoder | Decoder |
| --- | --- |
| Input $64 \times 64$ binary image | Input $\in \mathbb{R}^{10}$ |
| $4 \times 4$ conv. 32 ReLU. stride 2 | FC. 256 ReLU. |
| $4 \times 4$ conv. 32 ReLU. stride 2 | $4 \times 4$ upconv. 256 ReLU. stride 2 |
| $4 \times 4$ conv. 64 ReLU. stride 2 | $4 \times 4$ upconv. 64 ReLU. stride 2. |
| $4 \times 4$ conv. 64 ReLU. stride 2 | $4 \times 4$ upconv. 64 ReLU. stride 2 |
| $4 \times 4$ conv. 256 ReLU. stride 1 | $4 \times 4$ upconv. 32 ReLU. stride 2 |
| FC 256. FC. $2 \times 10$ | $4 \times 4$ upconv. 32 ReLU. stride 2 |

Table 4: Encoder and decoder architecture for disentangled representation learning on 3D Chairs.

| Encoder | Decoder |
| --- | --- |
| Input $64 \times 64 \times 3$ | Input $\in \mathbb{R}^{16}$ |
| $4 \times 4$ conv. 32 ReLU. stride 2 | FC. 256 ReLU. |
| $4 \times 4$ conv. 32 ReLU. stride 2 | $4 \times 4$ upconv. 256 ReLU. stride 2 |
| $4 \times 4$ conv. 64 ReLU. stride 2 | $4 \times 4$ upconv. 64 ReLU. stride 2. |
| $4 \times 4$ conv. 64 ReLU. stride 2 | $4 \times 4$ upconv. 64 ReLU. stride 2 |
| $4 \times 4$ conv. 256 ReLU. stride 1 | $4 \times 4$ upconv. 32 ReLU. stride 2 |
| FC 256. FC. $2 \times 10$ | $4 \times 4$ upconv. 32 ReLU. stride 2 |

### A.1    PI PARAMETER TUNING AND SET POINT GUIDELINES

We can tune PI parameters by following the conditions that guarantee the stability of the proposed method in Eq.(19) in Appendix C. In addition, $\beta(0)$ is initialized to a sufficiently large value in order to guarantee the KL-divergence is closed to zero at the beginning of model training. On the other hand, the choice of desired value of KL-divergence (set point) is very simple. Since our method achieves good disentanglement when $\beta = 1$, we can set its desired value to equal or lower than the KL-divergence of the original VAE as it converges.

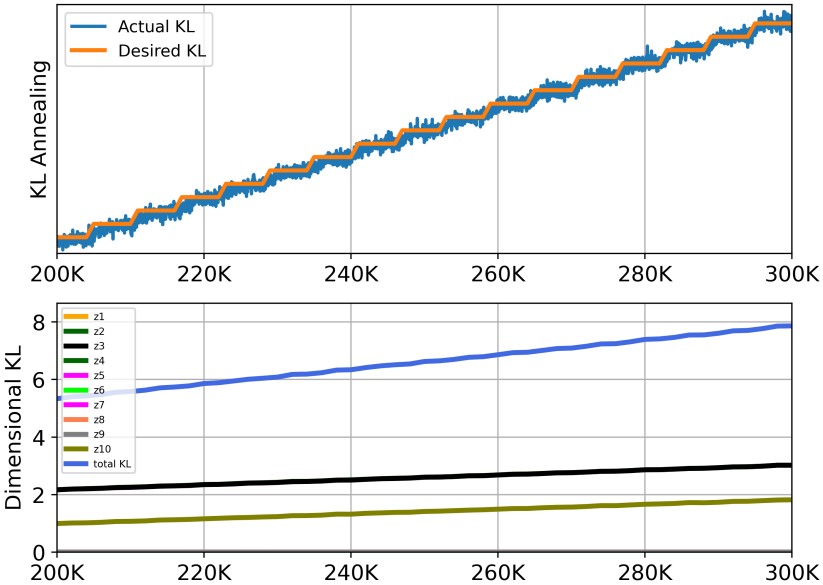

Figure 7: Top: A hybrid annealing method that combines step function with ramp function. Bottom: dimwise KL divergence.

---

**Algorithm 1** Incremental PI Control

---

1: **Input:** desired KL $C$, coefficients $K_p$, $K_i$, $\beta_{min}$, iterations $N$, window $T$
2: **Output:** weight $\beta(t)$ at training step $t$
3: **Initialization**: $\beta(0) = 150\ (100)$, $y_{KL}(0) = 0$
4: **for** $t = 1$ **to** $N$ **do**
5:     Sample KL-divergence, $y_{KL}(t)$
6:     $y(t) = \sum_{t-T}^{t} \alpha_i y_{KL}(i)$
7:     $e(t) \leftarrow C - y(t)$
8:     $dP(t) \leftarrow K_p[\sigma(-e(t)) - \sigma(-e(t-1))]$
9:     $dI(t) \leftarrow K_i e(t)$
10:     **if** $\beta(t-1) < \beta_{min}$ **then**
11:       $dI(t) \leftarrow 0$ // wind up
12:     **end if**
13:     $d\beta(t) \leftarrow dP(t) + dI(t)$
14:     $\beta(t) \leftarrow d\beta(t) + \beta(t-1)$
15:     **if** $\beta(t) < \beta_{min}$ **then**
16:       $\beta(t) \leftarrow \beta_{min}$
17:     **end if**
18:     **Return** $\beta(t)$
19: **end for**

---

## A.2 HYBRID ANNEALING METHOD

Fig. 7 shows the hybrid annealing method that combines step function with ramp function to gradually increase the desired value of KL-divergence for DynamicVAE.

## B ALGORITHM

We summarize the incremental PI algorithm in Algorithm 1.

## C PROOF OF STABILITY IN THEOREM 3.1

In this section we provide the omitted proof in the main paper about Theorem 3.1. For convenience purpose, we first restate Theorem 3.1 below.

**Theorem 3.1.** Let $a > 0$ and assume $g'(x) < 0, \forall x > 0$. Then DynamicVAE is stable at the equilibrium point $C$ if and only if the parameters of the PI controller, $K_i$ and $K_p$, satisfy the

following conditions

$$
\begin{cases}
K_p + K_i < -\dfrac{4(1+a)}{ag'(x_1^*)} \\
-0.5K_p^2 ag'(x_1^*)^2 - 2[K_p - 8K_i(1+a)]g'(x_1^*) + 8(1+a) > 0 \\
K_i > 0, K_p > 0
\end{cases}
\tag{19}
$$

*Proof.* At a high level, the proof goes by showing the spectral norm of the Jacobian matrix $A$ to be strictly less than 1 under the given condition, which is both sufficient and necessary for stability. To start with, recall that the Jacobian matrix $A$ at equilibrium point $x^*$ is defined by

$$
A = \begin{bmatrix} \frac{\partial f_1}{\partial x_1} & \frac{\partial f_1}{\partial x_2} & \frac{\partial f_1}{\partial x_3} \\ \frac{\partial f_2}{\partial x_1} & \frac{\partial f_2}{\partial x_2} & \frac{\partial f_2}{\partial x_3} \\ \frac{\partial f_3}{\partial x_1} & \frac{\partial f_3}{\partial x_2} & \frac{\partial f_3}{\partial x_3} \end{bmatrix}_{|x=x^*} = \begin{bmatrix} K_1 & K_2 & K_3 \\ K_4 & K_5 & 0 \\ 0 & 1 & 0 \end{bmatrix},
\tag{20}
$$

where

$$
K_1 = \frac{\partial f_1}{\partial x_1}|_{x_1=x_1^*} = 1
\tag{21a}
$$

$$
K_2 = \frac{\partial f_1}{\partial x_2}|_{x_2=x_2^*} = K_i + K_p\sigma(x_2^* - C)[1 - \sigma(x_2^* - C)] = \frac{1}{4}K_p + K_i
\tag{21b}
$$

$$
K_3 = \frac{\partial f_1}{\partial x_3}|_{x_3=x_3^*} = -K_p\sigma(x_3^* - C)[1 - \sigma(x_3^* - C)] = -\frac{1}{4}K_p
\tag{21c}
$$

$$
K_4 = \frac{\partial f_2}{\partial x_1}|_{x_1=x_1^*} = \frac{a}{1+a}g'(x_1^*)
\tag{21d}
$$

$$
K_5 = \frac{\partial f_2}{\partial x_2}|_{x_2=x_2^*} = \frac{1}{1+a}
\tag{21e}
$$

$$
\frac{\partial f_2}{\partial x_3}|_{x_3=x_3^*} = 0
\tag{21f}
$$

$$
\frac{\partial f_3}{\partial x_1}|_{x_1=x_1^*} = 0, \quad \frac{\partial f_3}{\partial x_2}|_{x_2=x_2^*} = 1, \quad \frac{\partial f_3}{\partial x_3}|_{x_3=x_3^*} = 0.
\tag{21g}
$$

In order to guarantee the stability of our state space model, the modulus of eigenvalue $\lambda$ of $A$ should be smaller than 1, i.e., $|\lambda| < 1$. By definition, the eigenvalues of $A$ can be obtained by computing the roots of the following characteristic polynomial:

$$
det(\lambda I - A) = \begin{bmatrix} \lambda - K_1 & -K_2 & -K_3 \\ -K_4 & \lambda - K_5 & 0 \\ 0 & -1 & \lambda \end{bmatrix}
$$
$$
= \lambda^3 - (K_1 + K_5)\lambda^2 + (K_1K_5 - K_2K_4)\lambda - K_3K_4 = 0
\tag{22}
$$

Instead of computing the (complex) roots of the above cubic polynomial analytically, we use the following bilinear transformation (Jury, 1964) to map the unit circle $|\lambda| < 1$ to the left half plane such that its real root is less than 0 (Hughes, 2015):

$$
\xi = \frac{\lambda - 1}{\lambda + 1} \quad \Longleftrightarrow \quad \lambda = -\frac{\xi + 1}{\xi - 1}.
\tag{23}
$$

Substituting $\lambda$ in Eq.(22) with (23), we have

$$
b_3\xi^3 + b_2\xi^2 + b_1\xi + b_0 = 0,
\tag{24}
$$

where

$$
\begin{cases}
b_3 = K_1 + K_5 + K_1K_5 - K_2K_4 + K_3K_4 + 1 \\
b_2 = K_1 + K_5 - K_1K_5 + K_2K_4 - 3K_3K_4 + 3 \\
b_1 = -K_1 - K_5 - K_1K_5 + K_2K_4 + 3K_3K_4 + 3 \\
b_0 = -K_1 - K_5 + K_1K_5 - K_2K_4 - K_3K_4 + 1
\end{cases}
\tag{25}
$$

Clearly, using the above transformation, we know that $|\lambda| < 1$ iff the real part of $\lambda$ is less than 0, i.e., $\text{Re}\{\xi\} < 0$. In order to ensure $\text{Re}\{\xi\} < 0$, based on Routh–Hurwitz stability criterion (Zabczyk,

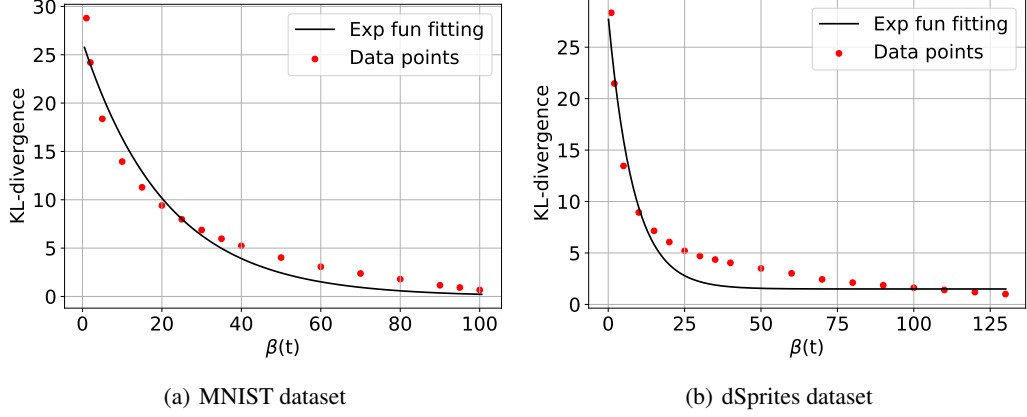

(a) MNIST dataset           (b) dSprites dataset

Figure 8: (a) $g(x(t))$ on MNIST dataset. (b) $g(x(t))$ on dSprites dataset.

1992), $b_0$, $b_1$, $b_2$, $b_3$ should satisfy the following sufficient and necessary condition.

$$
\begin{cases}
b_0 > 0 \\
b_1 > 0 \\
b_2 > 0 \\
b_1 b_2 > b_0 b_3
\end{cases}
\tag{26}
$$

Substitute Eqs. (21), (24) and (26) into the above system of inequalities, yielding

$$
\begin{cases}
b_3 = \dfrac{4a + 8 - (K_p + 2K_i)ag'(x_1^*)}{2(1+a)} > 0 \\[2mm]
b_2 = \dfrac{4(1+a) + (K_p + K_i)ag'(x_1^*)}{(1+a)} > 0 \\[2mm]
b_1 b_2 - b_3 b_0 = \dfrac{-0.5K_p^2 a^2 g'(x_1^*)^2 - 2a[K_p - 8K_i(1+a)]g'(x_1^*) + 8a(1+a)}{(1+a)^2} > 0 \\[2mm]
b_0 = \dfrac{-K_i a}{1+a} g'(x_1^*) > 0
\end{cases}
\tag{27}
$$

To complete the proof, recall that $a > 0$ and we assume $g'(x) < 0, \forall x$. Hence the coefficients of PI controller, $K_p$ and $K_i$ in Eq.(27), need to satisfy the following conditions.

$$
\begin{cases}
K_p + 2K_i > \dfrac{4(2+a)}{ag'(x_1^*)} \\[2mm]
K_p + K_i < -\dfrac{4(1+a)}{ag'(x_1^*)} \\[2mm]
- 0.5K_p^2 ag'(x_1^*)^2 - 2[K_p - 8K_i(1+a)]g'(x_1^*) + 8(1+a) > 0 \\[2mm]
K_i > 0
\end{cases}
\tag{28}
$$

Since $K_p > 0, K_i > 0$ in our designed PI control algorithm and $g'(x_1^*) < 0$, we can further simplify it as

$$
\begin{cases}
K_p + K_i < -\dfrac{4(1+a)}{ag'(x_1^*)} \\[2mm]
- 0.5K_p^2 ag'(x_1^*)^2 - 2[K_p - 8K_i(1+a)]g'(x_1^*) + 8(1+a) > 0 \\[2mm]
K_i > 0, K_p > 0
\end{cases}
\tag{29}
$$

Therefore, as $K_p$ and $K_i$ meet the above conditions (29), our DynamicVAE would be stable at the set point, which is verified by the following experiments on different datasets. ■

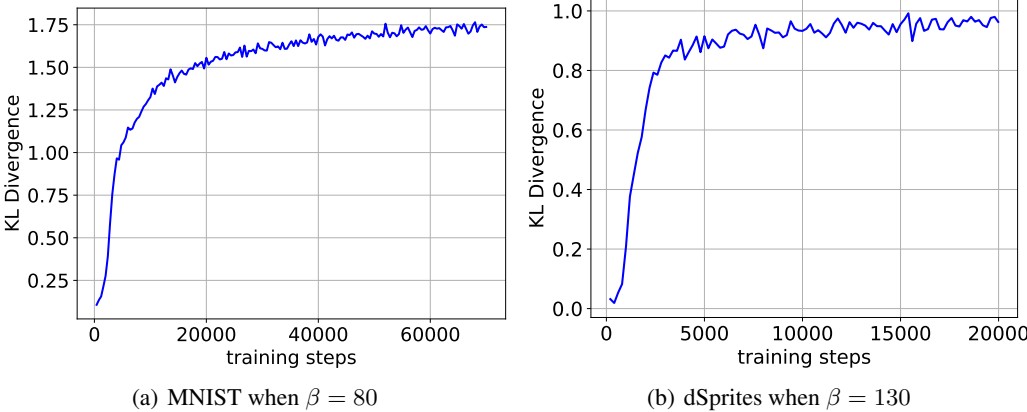

(a) MNIST when $\beta = 80$          (b) dSprites when $\beta = 130$

Figure 9: Time response of KL-divergence under different $\beta$ on MNIST and dSprites datasets respectively

## C.1    VERIFICATION ON BENCHMARK DATASETS

We first verify the validity of our assumption that $g'(x) < 0$ in Theorem 3.1. Fig. 8 illustrates the relationship between $\beta(t)$ and the actual KL when model training converges on dSprites and MNIST datasets. We can observe that the actual output KL-divergence and $\beta(t)$ have a highly negative correlation, which means $g'(x) < 0$.

Next, we are going to verify the stability of the proposed DynamicVAE on MNIST and dSprites datasets. On MNIST dataset, its mapping function $g(x)$ in Fig. 8 (a) can be approximately obtained by curve fitting with the following negative exponential function:

$$g(x(t)) = 26.38 \exp(-0.0476x(t)). \tag{30}$$

And the corresponding derivative is

$$g'(x(t)) = -1.26 \exp(-0.0476x(t)) \le -1.26. \tag{31}$$

In addition, we introduce how to obtain the hyperparameter $a$ in our dynamic model in Eq. (14). Assume that KL-divergence converges to a certain value $C'$ in the open loop control system during model training, then the dynamic model in Eq. (14) can be rewritten as

$$y(t) - y(t-1) + ay(t) = aC'. \tag{32}$$

When $y(0) = 0$, and the sampling period of our system is $T_s = 1$, the corresponding solution is given by

$$y(t) = C'(1 - \exp(-at)). \tag{33}$$

In order to obtain the value of $a$, one commonly used method in control theory is to set $a = \frac{1}{t^*}$ as we have $y(t^*) = C'(1 - \exp(-1)) \approx 0.632C'$. In this way, we can derive $a$ based on the training steps $t^*$ as KL-divergence reaches 63.2% of its final value $C'$ (Isa et al., 2011) in the experiments, as shown in Fig. 9. For MNIST dataset, we can get the hyperparameter $a = \frac{1}{5000}$ around based on the time response of KL-divergence in the open loop system, as shown in Fig. 9 (a).

Similarly, the derivative of mapping function on dSprites can be approximately expressed by

$$g'(x(t)) = -3.2 \exp(-0.121x(t)) \le -3.2. \tag{34}$$

In addition, we can get the hyperparameter $a = \frac{1}{2500}$ around based on the time response of KL-divergence in the open loop system, as shown in Fig. 9 (b).

We summarize the parameters $a$ and $g'(x(t))$ for different datasets in the following Table 5.

In this paper, we choose $K_p = 0.01$ and $K_i = 0.005$ with the parameters in Table 5 to validate our model meets the conditions in Eq. (29). In addition, our experimental results in Section 4 further demonstrate that our method can stabilize the KL-divergence to the set points.

Table 5: Parameters summary for different datasets

| Dataset | $a$ | $g'(x(t))_{min}$ |
|---------|-----|------------------|
| MNIST | $\frac{1}{5000}$ | -1.26 |
| dSprites | $\frac{1}{2500}$ | -3.2 |

# D  EXTRA EXPERIMENTS ON MNIST

## D.1  EVALUATION ON RECONSTRUCTION QUALITY

Fig. 10 shows the comparison of reconstruction loss and weight $\beta(t)$ for different methods. It can be observed that DynamicVAE and ControlVAE have comparable reconstruction accuracy to the basic VAE, but they have better disentanglement than it, as shown in Fig. 4 and 15. In addition, we can see that DynamicVAE has better reconstruction quality than the two $\beta$-VAE models.

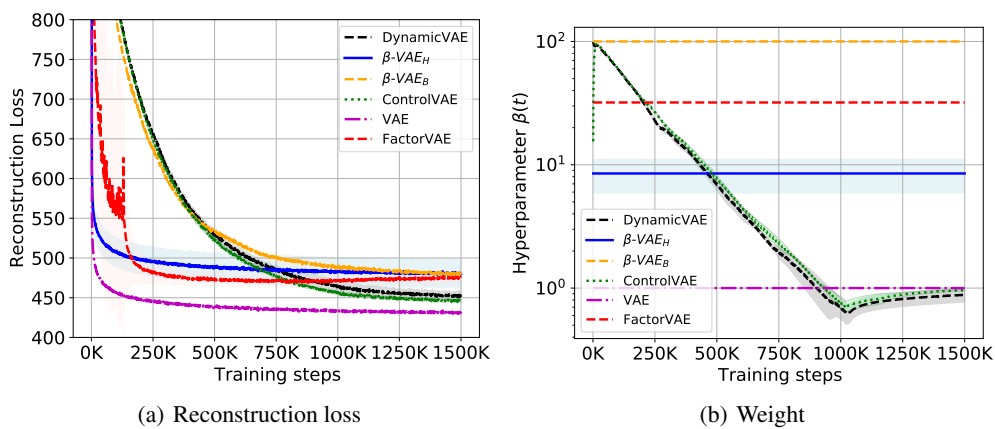

(a) Reconstruction loss      (b) Weight

Figure 10: Performance comparison of different methods.

## D.2  MNIST LATENT TRAVERSALS FOR BASELINES

We present some samples of latent traversals for the baseline methods. We find that DynamicVAE outperforms ControlVAE in term of rotation factor as illustrated in Fig. 4 and 11, though they have comparable disentanglement score. Moreover, DynamicVAE has a better disentanglement than $\beta$-VAE and FactorVAE in the following figures.

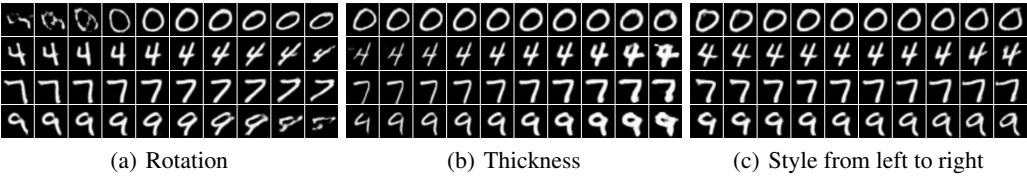

(a) Rotation      (b) Thickness      (c) Style from left to right

Figure 11: Latent traversals on MNIST for ControlVAE.

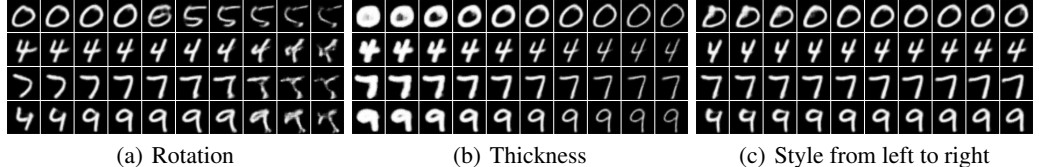

(a) Rotation            (b) Thickness            (c) Style from left to right

Figure 12: Latent traversals on MNIST for FactorVAE.

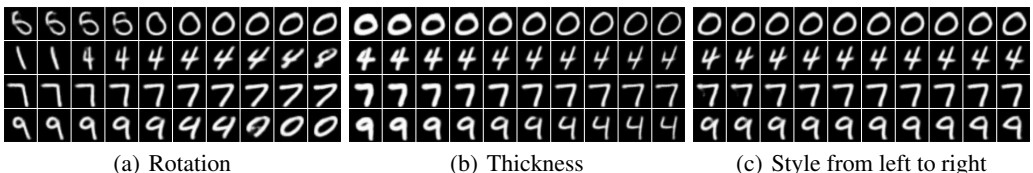

(a) Rotation            (b) Thickness            (c) Style from left to right

Figure 13: Latent traversals on MNIST for $\beta$-VAE$_H$ ($\beta = 10$).

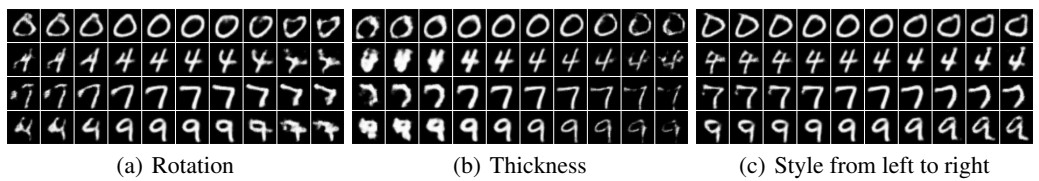

(a) Rotation            (b) Thickness            (c) Style from left to right

Figure 14: Latent traversals on MNIST for $\beta$-VAE$_B$ ($\gamma = 100$).

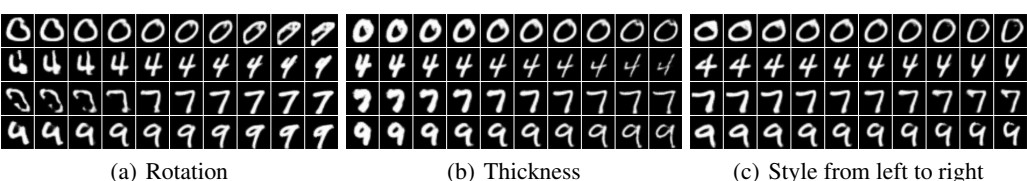

(a) Rotation            (b) Thickness            (c) Style from left to right

Figure 15: Latent traversals on MNIST for the basic VAE.

