# OpenReview forum: "DynamicVAE: Decoupling Reconstruction Error and Disentangled Representation Learning"
_ICLR.cc/2021/Conference — Reject_

### Official Review · AnonReviewer4 · 2020-10-22
**Contribution too small for an ICLR paper**

**Rating:** 4
**Confidence:** 4

**Review:**

This paper introduces a strategy for controlling the beta value of a beta-VAE during training using approaches from control theory that allow it to target a designated level of KL divergence between the encoder distribution and the prior.  This is done in a way that aims to achieve good reconstructions while maintaining disentangling performance.  It can be viewed as a refinement of the ControlVAE approach of Shao et al 2020, varying only in the specifics of the strategy used.

Though the approach is sensible and empirical results of the work are reasonable, I believe the contribution of the work is too small and too specific for publication at ICLR.  None of the high-level ideas are new, having already been introduced in Shao et al 2020, with the contribution effectively equating to making relatively low-level adjustments to the precise approach of that work.  This makes the content very niche and I believe it will be of limited interest to the ICLR community.   Beyond this, I also have serious concerns about the fact that this is an improvement on a somewhat outdated and discredited approach to disentanglement (the beta-VAE), along with the fact that the paper suffers from very serious overclaiming (such as the repeated incorrect claim of "removing the inherent trade-off between reconstruction accuracy and disentanglement).  I also have some misgivings with the experimental evaluation as quantitative assessment of disentanglement is only provided for one dataset and reconstruction comparisons for only two of the three datasets; not enough to reach any solid conclusions about the approaches' performance, particularly once we consider how inconsistent disentangling methods and metrics are known to be across random seeds (e.g. the gains over ControlVAE do not look statistically significant).

*Strengths*
- The paper is reasonably well written and easy to follow.
- The general approach is sensible and the control-theory approaches used to control beta are well-principled.
- The experimental results are reasonable, albeit not particularly impressive.


*Weaknesses*
- The contributions are very small and specific, constituting low-level changes relative to Shao et al 2020.
- The approach builds on the beta-VAE which is not only far from the state-of-the-art, but actively discredited in various ways (e.g. Mathieu et al 2019 and Locatello et al 2019).  Though overwise quite well-referenced, the paper critically omits any discussion of these limitations and I feel like the community has somewhat moved on from the beta-VAE by now; I do not think this is a line of work which will last the test of time or which has any particularly useful transferable ideas.  Given that unsupervised disentanglement methods have rather limited applications at the moment, I thus do not feel the work adds much to the community.
- The paper severely overclaims: it results suggest that it is potentially able to improve on the Pareto front between reconstruction and disentanglement, but it is blatantly untrue that it completely removes the trade-off in the way that is repeatedly claimed.  Firstly, one still has to have a target KL and how this is fixed naturally induces a trade-off (with higher targets being more focused on disentanglement).  Secondly, there are various reasons why there will always be a degree of trade-off between these objectives regardless of the approach used (e.g. the best reconstructions are achieved if the autoencoder is completely noiseless, but removing the noise also removes any pressure for the latent space to structure in any way as one can just construct a lookup table), such that the claims clearly cannot hold.
- As explained earlier, the numerical experimental results are not really extensive enough to confirm the approach offers meaningful improvements across a range of tasks.  For example, disentanglement is only quantitively evaluated on one dataset (Dsprites) and even more reruns are needed because the differences to the ControlVAE do not look like they are statistically significant.
- I think there are much stronger baselines that could have been considered such as a) disentangling approaches published since the FactorVAE, b) using hierarchical VAEs which has shown to already give a much better trade-off (see eg arXiv:1906.00230), c) more heuristical ways of varying beta, and d) a FactorVAE where the scaling on the total correlation term is varied.  Relatedly, the FactorVAE is still rapidly improving its reconstruction loss with more iterations in Fig 2a, making the comparisons a little unfair as it seems like it might quickly catch up if the training was run for longer.

---

### Official Review · AnonReviewer3 · 2020-10-28
**Limited empirical evaluation.**

**Rating:** 4
**Confidence:** 5

**Review:**

#### Summary
The paper proposes DynamicVAE, a variant of ControlVAE that is claimed to decouple the optimization reconstruction error and disentanglement, hence mitigating the inherent trade-off between the two.
#### Pros:
- The paper sheds light on the tradeoff problem in disentangled representation learning from an optimization perspective.
- They empirically show that their proposed method can in fact decouple the two optimization objectives.

#### Cons:
- It will be very helpful if the authors can clarify the exact difference between DynamicVAE and ControlVAE. My understanding is that the main difference lies in their usage of Hybrid Annealing, which is certainly important but somewhat incremental given the overall results.
- My main concern with the paper is that it lacks extensive empirical evaluation.
  1. Currently, the baselines used are very week. Please consider using baseline methods such as [1] or others that also tackle the trade-off problem.
  2. Please consider using more challenging datasets like SmallNorb or Cars (see [2] for options). This is important since dSprites can be fully specified using only the disentangled factors. For these difficult datasets please consider doing a quantitative evaluation. Currently, I only see qualitative evaluation for Chairs (which indeed is a reasonable dataset).
  3. What values of beta and gamma did you try for the baseline methods? The RMIG numbers reported for beta vae are very low and different from what RMIG paper reports.

#### Comments:
- It is known that the pixel-wise reconstruction score does not give a clear picture of the sample quality [3]. Please consider using FID (see [4]).
- I am interested in knowing how the high KL of 20 affects the NLL of the model? Also, how is the sample quality of generated samples?

My current rating is primarily based on the fact that the paper requires a lot more evaluation with competitive baselines on more relevant datasets. Also, clarifying the difference from ControlVAE will really help.

[1] Overcoming the Disentanglement vs Reconstruction Trade-off via Jacobian Supervision
[2] Challenging Common Assumptions in the Unsupervised Learning of Disentangled Representations
[3] Generating Diverse High-Fidelity Images with VQ-VAE-2
[4] Improving the Reconstruction of Disentangled Representation Learners via Multi-Stage Modelling

---

### Official Review · AnonReviewer1 · 2020-10-28
**I am not convinced that the proposed method has sufficient novelty. More experimental results might be necessary.**

**Rating:** 4
**Confidence:** 4

**Review:**

In $\beta$-VAE, one challenge is to choose the hyper-parameter $\beta$ that controls the trade-off between the reconstruction quality and the disentanglement. This paper proposes a method called DynamicVAE. Rather than using a fixed hyperparameter $\beta$, the method leverages a modified incremental Proportional-integral (PI) controller, which dynamically tunes $\beta$ at different stages of training. The method is tested on benchmark datasets.

The paper is not difficult to follow. The idea of dynamically tuning the hyper-parameter looks interesting. However, there exists a previous method called control-VAE, which also dynamically tunes $\beta$ with a PI controller. Although it is reported in the paper that the proposed method outperforms controlVAE, it is not clear to me how it differs from it. Therefore, I am not sure whether the proposed method has sufficient novelty.

As shown in Table 1, DynamicVAE outperforms FactorVAE, but the difference is tiny. In particular, Dynamic VAE is better in disentangling shape and scale, while FactorVAE is better in disentangling pos. x, pos. y and orientation. The results make me doubt whether DynamicVAE consistently outperforms FactorVAE in various datasets. It would make the paper more convincing if the authors can report quantitative results on more datasets.

The authors claim that the proposed method decouples reconstruction and disentanglement. However, no quantitative measurements for reconstruction are reported. Therefore, it is not clear whether the proposed method outperforms the baselines in terms of reconstruction.

In summary, I do not suggest accepting this paper. I am not convinced that the proposed method has sufficient novelty. More experimental results might be necessary.

==============================================================================================

Thanks for the authors' response. I am still inclined to reject this paper. Compared to the existing ControlVAE, the contributions of this paper looks incremental. More experimental results might be necessary to make the paper more convincing.

---

### Official Review · AnonReviewer2 · 2020-11-08
**Dynamically controlling the beta parameter is an interesting idea but this paper has several shortcomings and is not ready for publication.**

**Rating:** 4
**Confidence:** 5

**Review:**

Positive points:

- Dynamically controlling the beta parameter of beta VAE is an interesting idea

Negative points:

- 1. The authors do not put their work into the context of the closely related information bottleneck principle
- 2. The authors do not provide further details on several mechansims of their proposed method:
  - 2. a) How is the constraint threshold C on the KL divergence term chosen? Does it have to be tuned for every dataset? Can it be tuned at all in the unsupervised setting, when no information about the ground truth factors is available?
  - 2. a) Why does a constraint on the KL divergence result in a decoupling of reconstruction quality and disentanglement?
  - 2. b) Is the beta schedule of starting with a large beta value which is then reduced a result of the control algorithm or is it pre-defined by the Hybrid annealing method the authors describe in section 3?
- 3. Experiments:
  - 3. a) FactorVAE has not converged yet, and it is the closest competitor.
  - 3. b) The RMIG scores of FactorVAE, ControlVAE and DynamicVAE reported in Table 1 are not significantly different and therefore the score of DynamicVAE should not be in bold font.
- 4. Missing references


1. Missing references and context to related work:

The method seems closely related to the information bottleneck method (Tishby et al. 2000, Kirsch et al. 2020): the constraint on the KL divergence term to remain below a fixed value, defines an upper bound on the bandwith of the latent channel. The authors should explore this connection further, or at least should discuss the relation between the proposed method and the information bottleneck method.

(Stuehmer et al. 2020) et al. propose a structured prior that also reduces the trade-off between reconstruction accuracy and disentanglement. It would be interesting to compare the performance of the proposed method to their methood ISA-VAE.


2. Method

2. a) How is the constraint threshold C on the KL divergence term chosen? Does it have to be tuned for every dataset? Can it be tuned at all in the unsupervised setting, when now information about the ground truth factors is available?

2. b) Is the beta schedule of starting with a large beta value which is then reduced a result of the control algorithm or is it pre-defined by the Hybrid annealing method the authors describe in section 3?


3. Experiments

Fig. 2 a) depicts the reconstruction error of the proposed method and different baselines for up to 1250K iterations. Figure 6 depicts the reconstruction loss of the the proposed method for up to 1600K iterations.

3. a) FactorVAE has not converged yet (Fig. 2 a), and it is the closest competitor. Further iterations are recquired to assess if there is a difference in the reconstruction error between the proposed method and FactorVAE.

3. b) The RMIG scores of FactorVAE, ControlVAE and DynamicVAE reported in Table 1 are not significantly different and therefore the score of DynamicVAE should not be in bold font. Are the reported error values standard deviation? This needs to be explained in the caption of the table.

3. c) How does the proposed PI-controller based approach compare to a simpler approach of beta-VAE with a fixed beta schedule: After a pre-defined number of iterations beta goes down to a lower value. Does the dynamic control have any benefits?

3. d) Related to 3 c): Is the schedule for the beta parameter that is generated by the PI-controller significantly different for different datasets?


Conclusion: Dynamically controlling the beta parameter is an interesting idea but this paper has several shortcomings and is not ready for publication.


References:

Tishby et al., The information bottleneck method, ACCCC 1999
Kirsch et al., Unpacking Information Bottlenecks: Unifying Information-Theoretic Objectives in Deep Learning, arXiv 2020
Stuehmer et al., Independent Subspace Analysis for Unsupervised Learning of Disentangled Representations, ICML 2020

---

### Decision · Program_Chairs · 2021-01-07
**Final Decision**

**Decision:**

Reject

**Comment:**

The paper is in general well written and easy to follow, and the considered approach of controlling beta is sensible. However, all reviewers identify shortcomings in the empirical analysis of the proposed method (missing comparison with stronger baselines, convergence issues of the considered baselines, considered datasets, etc.). Furthermore, compared to the ControlVAE the contribution of the paper seems limited; and the empirical evaluation is insufficient to claim superior results in general. The authors did not address most of the concerns raised by the reviewers in their rebuttal. The authors can improve their paper substantially by performing the experimental results proposed by the reviewers and clarifying differences to the ControlVAE—but in its current form the paper does not meet the standard of ICLR.